# Relationships between Participation in Volunteer-Managed Exercises, Distance to Exercise Facilities, and Interpersonal Social Networks in Older Adults: A Cross-Sectional Study in Japan

**DOI:** 10.3390/ijerph182211944

**Published:** 2021-11-13

**Authors:** Yuki Soma, Ayane Sato, Kenji Tsunoda, Naruki Kitano, Takashi Jindo, Takumi Abe, Tomohiro Okura

**Affiliations:** 1Faculty of Education, Hirosaki University, 1 Bunkyo-cho, Aomori 036-8560, Japan; 2Faculty of Regional Collaboration, Kochi University, 2-5-1 Akebono-cho, Kochi 780-8520, Japan; sato.ayane@kochi-u.ac.jp; 3Faculty of Social Welfare, Yamaguchi Prefectural University, 3-2-1 Sakurabatake, Yamaguchi 753-8502, Japan; ktsunoda@yamaguchi-pu.ac.jp; 4Physical Fitness Research Institute, Meiji Yasuda Life Foundation of Health and Welfare, 150 Tobuki, Tokyo 192-0001, Japan; na-kitano@my-zaidan.or.jp; 5Faculty of Health and Sport Sciences, University of Tsukuba, 1-1-1 Tennodai, Ibaraki 305-8574, Japan; jindo.takashi.ge@u.tsukuba.ac.jp (T.J.); okura.tomohiro.gp@u.tsukuba.ac.jp (T.O.); 6Integrated Research Initiative for Living Well with Dementia, Tokyo Metropolitan Institute of Gerontology, 35-2 Sakae, Tokyo 173-0015, Japan; abe@tmig.or.jp; 7Centre for Urban Transitions, Swinburne University of Technology, Hawthorn, VIC 3122, Australia

**Keywords:** participation, interpersonal social networks, accessibility, volunteer-managed exercises, older adults

## Abstract

This study aimed to examine the factors related to participation in volunteer-managed preventive care exercises by focusing on the distance to exercise facilities and interpersonal social networks. A postal mail survey was conducted in 2013 in Kasama City in a rural region of Japan. Older adults (aged ≥ 65 years) who were living independently (*n* = 16,870) were targeted. Potential participants who were aware of silver-rehabili taisou exercise (SRTE) and/or square-stepping exercise (SSE) were included in the analysis (*n* = 4005). A multiple logistic regression analysis revealed that social and environmental factors were associated with participation in SRTE and SSE. After adjusting for confounding variables, exercise participation was negatively associated with an extensive distance from an exercise facility in both sexes for SRTE and SSE. Among women, participation in SRTE was negatively associated with weak interpersonal social networks (odds ratio (OR) = 0.57), and participation in SRTE and SSE was negatively associated with being a car passenger (SRTE, OR = 0.76; SSE, OR = 0.60). However, there were no significant interactions between sex and social and environmental factors. Our findings suggest the importance of considering location and transportation to promote participation in preventive care exercise.

## 1. Introduction

Physical activity provides various health benefits [1,2,3], and recently, it has been found that volunteer-managed exercise programmes can improve older people’s quality of life in terms of physical domains and self-rated health [4,5]. Older adults can gain considerable benefits from volunteer-managed exercise programmes and professionally supervised programmes [6]. In Japan’s long-term care system, preventive care volunteers play an important role in providing preventive care activities (e.g., exercise programmes) for community-dwelling older adults [7]. Generally, these volunteers have been trained by local governments for this purpose. In recent years, the effectiveness of volunteer-managed exercise instruction has been reported in the United States, the United Kingdom, and Australia. Hence, such programmes are beginning to gather attention [8]. Volunteer-managed exercise programmes are expected to contribute to the well-being of older adults all over the world.

Although volunteer-managed activities can improve older adults’ health status, participation might be influenced by physical distance and interpersonal social networks. Previous studies have reported that traveling long distances to attend facilities inhibits the attendance of regular health check-ups and participation in salon activities [9,10]. Moreover, the individuals’ normal mode of transportation influences participation in various leisure and social activities [11,12]. On the other hand, the presence of rich interpersonal social networks (e.g., contact with friends) promotes primary care access among urban older adults [13]. These studies suggest that community-dwellers with long distances to facilities and weaker interpersonal social networks may be disadvantaged in terms of participation in volunteer-managed activities.

Although research on participation in volunteer-managed activities is required to enhance the participation rate, no studies have examined the relationships between participation, the distance to facilities, and interpersonal social networks in older adults. Furthermore, there is a gap between participation and awareness of volunteer-managed activities. To participate in an activity, there are two steps: first, becoming aware of the activity, and second, participating in the activity. Our previous study revealed that short distances between home and the nearest exercise facility, and rich interpersonal social networks, promote awareness of volunteer-managed preventive care exercises among community-dwelling older adults belonging to both sexes [14]. To promote the participation of older adults with awareness, research on them is needed. This type of information is beneficial for enhancing the participation rate in volunteer-managed exercise programmes. Additionally, since there are fundamental gender differences in physical fitness [15], types of physical activity participation [16], and types of social participation [17], there are likely to be gender differences in the factors of participation as well. However, there are no reports of gender differences being examined and stratified analyses being conducted in studies of accessibility.

This study aimed to examine factors related to older adults’ participation in certain types of volunteer-managed preventive care exercises by focusing on the distance to exercise facilities and interpersonal social networks. We hypothesised that weak interpersonal social networks and long distances to facilities would inhibit participation in these exercises, and that there would be gender-based differences.

## 2. Materials and Methods

### 2.1. Data

This cross-sectional study used inventory survey data that were collected in 2013 in Kasama City (population 78,279; area 240.3 km^2^), which is in a rural region in Ibaraki Prefecture, Japan. Questionnaires were mailed to 16,870 older people (aged ≥ 65 years) who were not certified as having care needs; 10,339 responses were received. The respondents’ ages ranged from 65 to 101 years. Of these responses, certain medical histories (i.e., stroke, dementia, mental disorders, exercise prohibited by a medical doctor, and non-responses regarding the medical history items, *n* = 1346), problems identifying their precise address (*n* = 422), or missing data in the analysed variables (*n* = 1873) were excluded from the analysis. The data of the remaining 6698 respondents (3197 men; 3501 women) were analysed. The participants were asked to read the explanation and provide informed consent in a consent cover letter, which included an explanation of the aim and procedure of the study. The University of Tsukuba’s ethical committee approved this study protocol (tai 26–31).

### 2.2. Dependent Variables

Two types of exercise—the silver-rehabili taisou exercise (SRTE) and square-stepping exercise (SSE)—were used as the preventive care exercises. These exercises have gradually spread in Kasama City; in 2013, 34 and 21 places offered SRTE and SSE, respectively. Both were provided by community-dwelling volunteers who regularly worked (i.e., once a week or once a month) at a community centre or similar facility. Volunteers were residents of the community, and formal training by doctors, physical trainers, and/or academic staff (university teachers, postgraduate students, etc.) was needed to teach both exercises.

SRTE is a preventive care exercise programme for older adults that began as a health promotion project initiated by Ibaraki Prefecture in 2005. It aims to improve older adults’ range of motion and mobility (standing, sitting, walking, etc.). SSE was created by a research group [18] and involves stepping in various directions while standing on a thin mat. A systematic review and meta-analysis reported that SSE can improve balance and prevent the fear of falling in older people [19]. The volunteer-managed SSE activity began in Kasama City in 2008. According to the differences in the characteristics of the exercise, SRTE is for older people with lower physical function levels compared to SSE.

The surveyed individuals’ participation in SRTE or SSE was defined using a question regarding awareness. The response options were: ‘I have done it’, ‘I know about it but have not done it’, or ‘I do not know about it’. Those who answered ‘I have done it’ were defined as participants. The characteristics of the participants in these exercises are shown in Appendix A.

### 2.3. Independent Variables

The independent variables included the distance from home to the exercise facility, the main mode of travel, and interpersonal social networks. To assess the distance to SRTE, we evaluated the road distances from the participants’ homes to the nearest exercise facility that conducted SRTE. The same procedure was performed for the SSE facilities to assess the distance to SSE. The main modes of travel (i.e., the participants driving a car themselves (driving), a car driven by another person (passenger), and cycling or walking) were assessed to identify the ease of long-distance travel.

The ArcGIS software (ArcGIS 10.3; Esri, Redlands, CA, USA) was used to calculate the road distance. The ArcGIS Data Collection Standard Pack (2014; Esri Japan, Tokyo, Japan) was used to obtain road line data. The Yahoo Geocoding API with the ‘RCurl (Ver. 1.98-1.3)’ and ‘RJSONIO (Ver. 1.3-1.4)’ packages in R software (Ver. 4.0.5) (R Foundation for Statistical Computing, Vienna, Austria) and Google Maps (https://www.google.co.jp/maps/, accessed on 13 November 2021) were used for geocoding. The accuracy of the respondents’ addresses and the locations of each exercise facility were set at the chiban, koban, and edaban levels (approximate location of the building).

For the main mode of travel, respondents selected the option most applicable to their daily life: ‘driving a car or motorcycle myself’, ‘car driven by another person’, or ‘cycling or walking’.

The interpersonal social networks were assessed via three questions. Two questions were cited in the Kihon Checklist [20]: ‘Do you visit a friend’s home?’ and ‘Do you have someone with whom you can talk when in trouble?’ One question was constructed specifically for this survey: ‘Do you listen to friends or family members who confide in you?’ [21]. The response options were ‘yes’ (1 point) or ‘no’ (0 points). The interpersonal social network total score was calculated by summing these items’ responses. We divided them into two categories: having a weak social network (0–1 points) or not (2–3 points).

### 2.4. Covariates

To adjust for potential confounding variables, we gathered data on age (65–74/75+ years old), education level (compulsory/senior high school or higher), living arrangements (living alone/not living alone), subjective economic status (poor/normal/good), arthralgia or neuralgia, estimated population density within a 1 km radius from the respondents’ residence, and participation in the other exercise focused on in this study.

The population density was estimated using the ArcGIS software and Japanese census data for 2010 from e-Stat [22,23]. The estimated neighbourhood population density within a 1 km radius was divided into tertile groups for each analysis, which were included as dummy variables.

### 2.5. Statistical Analysis

The medians with 25–75 percentiles were calculated for the continuous variables, and the proportions were calculated for the categorical variables. We used a Wilcoxon rank sum test, chi-square test, and Fisher’s exact test to compare participants and non-participants on the variables. Logistic regression analysis was used to examine the associations between participation and the independent variables after adjusting for the covariates. Considering the aforementioned gender and exercise characteristics, these univariate and multivariate analyses were stratified by the respondents’ sex and types of exercise. The odds ratios (OR) of the logistic regression analyses were graphically expressed using cubic spline curves to indicate the association between participation and each facility’s distance. When drawing the spline curve, an upper limit of each facility’s distance was set to 6000 m, and those who exceeded this limit were converted into upper values. In addition, gender differences in the association between participation, the distance to the exercise facility, and interpersonal social networks were examined using an interaction term. A *p*-value < 0.05 was considered statistically significant. All analyses were performed using Stata version 16.1 (Stata Corp, College Station, TX, USA).

## 3. Results

### 3.1. Respondents’ Characteristics

Of those who were included in the analysis, 4005 respondents (59.8%, aged 65 to 95 years) were aware of either SRTE or SSE. Of these respondents, those who were not aware of SRTE were excluded from the analysis of participation in SRTE (*n* = 95), and those who were not aware of SSE were excluded from the analysis of participation in SSE (*n* = 2002), resulting in a final analysis of 3910 and 2003 respondents, respectively. Table 1 shows the individual characteristics of the participants who did and did not participate in the exercises. The SRTE participation rate was 28.1% (men: 20.3%; women: 33%) and the SSE rate was 28.2% (men: 16.2%; women: 33.9%). The characteristics of the participants in these exercises are shown in Appendix A. SRTE participants were mostly male, older, and nearer exercise facilities and had a lower education level and a lower rate of driving a car.

The participants in SRTE were comparatively older and aware of the other exercise than non-participants for both sexes. The male participants had a significantly higher education level. Meanwhile, the female participants had a significantly higher rate of arthralgia or neuralgia, a higher rate of driving a car or motorcycle, and a lower rate of weak interpersonal social networks. Of the SSE group, the female participants had a higher rate of driving a car or motorcycle than the female non-participants.

SRTE participants showed a higher rate of participation in the other exercise than non-SRTE participants, and the same tendency was observed among SSE and non-SSE participants in both sexes. The facility’s distance was only related to women’s participation.

### 3.2. Participation Factors

After adjusting for the confounding variables, using a car driven by another person (OR = 0.76, 95% CI 0.59–0.98) relative to driving a car and weak interpersonal social networks (OR = 0.57, 95% CI 0.35–0.91) in women were significant, negatively associated factors (Table 2). For factors related to SSE (Table 3), participation in women was significantly negatively associated with using a car driven by another person (OR = 0.60, 95% CI 0.43–0.84). A distance of more than 500 m to the facility in women was a significant, negatively associated factor for participation in SRTE and SSE. Additionally, there was a partially negative association between the distance to SRTE and SSE facilities in men.

Figure 1 and Figure 2 show the L-shaped cubic spline curves. Women’s participation status declined sharply up to 2000 m and then declined slowly or plateaued for distance. This indicates that the decrease in OR associated with distance lessened with the increasing distance.

Table 4 shows the interaction between sex, main mode of travel, interpersonal social networks, and distance (>500 m, the distance at which the OR begins to show less than 1.0 in both exercises). There was no significant interaction between sex and social and environmental factors.

## 4. Discussion

This study examined the relationships between participation in volunteer-managed preventive care exercises that participants were aware of, interpersonal social networks, and the distance to an exercise facility in community-dwelling older adults living in Japan. The results reveal that distant exercise facilities inhibited women’s participation, but the association was limited in men. Interpersonal social networks were related to participation for women only in SRTE. These findings partially support our hypothesis.

Many studies have reported relationships between distance to health-related facilities and health behaviours. A systematic review showed that the distance to facilities and open green spaces may positively influence older adults’ physical activity [24,25]. Furthermore, previous studies have found a linear relationship between positive health behaviours and the distance to parklands, squares, and primary care services [9,26]. The present study found an L-shaped association between the participation rate and objectively measured road distance and a decrease in the participation rate as the distance increased in both sexes and exercises. In particular, the facility’s distance was a significant inhibiting factor for both types of exercise in older women. Additionally, a car driven by another person was only an inhibiting factor in the case of women. These results suggest that the distance from home to the exercise facility and the main mode of travel were important, robust factors for older women’s participation in volunteer-managed preventive care exercises, but that there were no notable sex differences. Driving a car has been reported to be a promoting factor for participation in social activities in participants living in urban areas [27]. Our results corroborate this finding. The reason that a significant difference was observed only for women may be because women use cars less often than men [28,29] and have a smaller activity area. The fact that public transportation is also less popular in rural areas [30], such as the area where this study was conducted, is considered to strengthen this effect.

Other research has shown that richer interpersonal social networks can promote participation in primary care and social activities. One study found that a richer interpersonal social network was associated with greater social participation in older adults [31]. Another study reported that primary care use was influenced by social cohesion [13]. Furthermore, older women with a larger interpersonal social network of non-relatives participated more in recreational activities, including hobbies and sports [32]. In our study, the only association between participation and interpersonal social networks was found in SRTE in women, and there was no interaction with sex. This suggests that the association between interpersonal social networks and participation in volunteer-managed preventive care exercises was not robust and that significant gender differences were not found. Some systematic reviews have also shown that, while social networks and social support are effective for social participation, they are not related [12]. The impact of social interaction on participation is likely to depend on the characteristics of the activity.

There were some limitations to our study. First, since this study was cross-sectional, we could not identify a causal association. Second, the broad generalisability of the findings was limited because this study was conducted in a middle-sized Japanese city and focused only on SRTE and SSE. Additional research is needed in cities of different sizes, with different characteristics and a variety of volunteer-managed activities. Third, participation in volunteer-managed exercises was judged binomially as participation or no participation; therefore, we could not obtain information concerning the type of participation (e.g., if participants belonged to an exercise group or if it was a temporary experience). Fourth, the study included participants who were aware of SRTE and SSE but did not survey their awareness of and participation in other types of exercise. Therefore, it is possible that there may be a selection or information bias. Fifth, this study analysed 2013 survey data, which may not be applicable to the current situation of the elderly. Finally, our survey contained only three items related to social networks, which might not have sufficiently assessed the social network aspect. Despite these limitations, this study has the advantage of objectively evaluating the distances to the facilities and collecting survey data. Our results provide strong evidence for promoting volunteer-managed preventive care exercises in local communities.

## 5. Conclusions

In conclusion, women’s participation in preventive care exercises (SRTE and SSE) was negatively associated with a longer distance to a facility. The robustness of a longer distance to a facility and a car driven by another person, in women, was supported. An effective strategy to promote participation in preventative care exercises would be to offer exercise groups in various locations and ensure convenient transportation.

## Figures and Tables

**Figure 1 ijerph-18-11944-f001:**
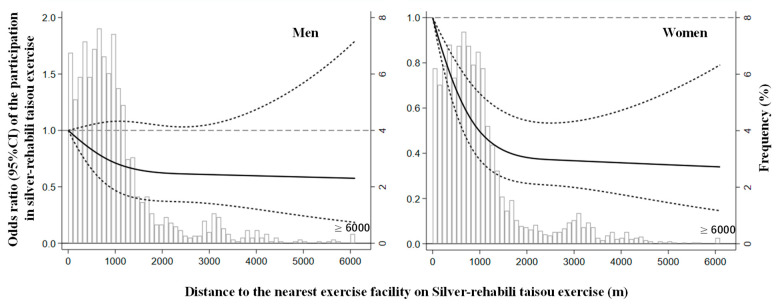
Odds ratio of participation in the silver-rehabili taisou exercise and the distance to a facility in men and women. The solid line represents the odds ratio, and the dashed line represents the 95% confidence intervals. Distance to the nearest exercise facility on silver-rehabili taisou exercise distribution is represented by histograms in the background. Knots were placed at the 10th, 50th, and 90th percentiles of the distance to the exercise facility using <100 m as the reference point. CI = confidence interval.

**Figure 2 ijerph-18-11944-f002:**
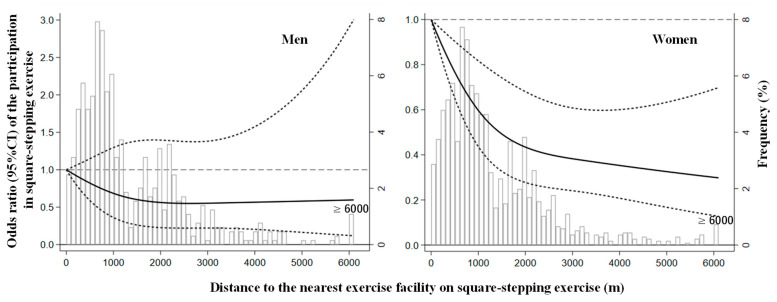
Odds ratio of participation in the square-stepping exercise and distance to a facility in men and women. The solid line represents the odds ratio, and the dashed line represents the 95% confidence intervals. Distance to the nearest exercise facility on square-stepping exercise distribution is represented by histograms in the background. Knots were placed at the 10th, 50th, and 90th percentiles of the distance to the exercise facility using <100 m as the reference point. CI = confidence interval.

**Table 1 ijerph-18-11944-t001:** Characteristics of the respondents.

		Silver-Rehabili Taisou Exercise				Square-Stepping Exercise			
		Men		Women		Men		Women	
		Participants	Non-Participants	Participants	Non-Participants	Participants	Non-Participants	Participants	Non-Participants
		*n* = 306	*n* = 1204	*n* = 792	*n* = 1608	*n* = 104	*n* = 539	*n* = 461	*n* = 899
Age (years)	Continuous	74 (70–79)	73 * (69–78)	72 (69–77)	72 * (68–77)	72 (69–78)	73 (69–77)	71 (68–75)	71 (68–76)
	65–74	51.6	59.2 *	60.2	65.6 *	57.7	60.9	71.6	67.0
	75+	48.3	40.8	39.8	34.4	42.3	39.1	28.4	33.0
Senior high school or higher		76.8	67.5 *	72.6	70.1	79.8	72.4	79.0	76.5
Living alone		8.8	6.6	16.8	14.6	8.7	6.7	16.7	13.2
Subjective economic status	Poor	13.7	16.5	11.2	13.6	13.5	13.2	9.3	11.3
	Normal	75.5	73.9	76.0	75.6	75.0	76.8	79.2	75.5
	Good	10.8	9.6	12.8	10.8	11.5	10.0	11.5	13.1
Arthralgia or neuralgia (yes)		19.3	16.8	29.5	25.4 *	21.2	16.3	26.2	25.1
Population density (*n*/km^2^)		1245.2 (436.6–1694.3)	1186.2 (412.7–1656.4)	1240.6 (462.2–1663.8)	1244.0 (462.6–1686.2)	1267.1 (456.5–1769.3)	1269.0 (458.3–1632.8)	1341.2 (566.5–1685.6)	1273.0 (552.5–1679.9)
Aware of the other exercise (yes)		52.9	36.5 *	69.6	47.0 *	92.3	93.9	96.7	95.7
Participation in the other exercise (yes)		20.9	2.7 *	37.9	9.1 *	61.5	18.2 *	65.1	27.9 *
Main mode of travel	Car (driving)	83.0	85.6	51.5	46.6 *	90.4	86.3	58.1	50.0 *
	Car (passenger)	4.2	3.7	20.1	24.3	3.8	2.6	14.1	21.0
	Cycling or walking	12.8	10.7	28.4	29.1	5.8	11.1	27.8	29.0
Weak interpersonal social networks		5.9	9.1	3.3	6.0 *	5.8	8.3	3.3	5.0
Distance to exercise facility (m)		880.0 (453.0–1280.3)	869.2 (506.3–1282.9)	785.2 (435.3–1193.5)	900.6 * (548.9–1327.6)	950.3 (575.2–2021.6)	1022.3 (604.3–2012.6)	948.6 (488.3–1666.4)	1065.3 * (655.5–1954.6)

Note. Values are median (25th, 75th percentile) or percentage. The *p*-value was obtained using the Wilcoxon rank sum test, chi-square test, or Fisher’s exact test between participants and non-participants. * *p* < 0.05.

**Table 2 ijerph-18-11944-t002:** Associations between participation in the silver-rehabili taisou exercise, interpersonal social networks, and distance to exercise facility.

		Men		Women	
		OR	95% CI	OR	95% CI
Main mode of travel	Car (driving)	1.00		1.00	
	Car (passenger)	1.10	[0.56, 2.19]	0.76	[0.59, 0.98] *
	Cycling or walking	1.41	[0.93, 2.12]	0.84	[0.66, 1.06]
Weak interpersonal social networks	No	1.00		1.00	
	Yes	0.68	[0.39, 1.17]	0.57	[0.35, 0.91] *
Distance to exercise facility	≤500 m	1.00		1.00	
	−1000 m	0.73	[0.51, 1.04]	0.65	[0.52, 0.83] *
	−1500 m	0.90	[0.62, 1.30]	0.64	[0.50, 0.84] *
	−2000 m	0.93	[0.52, 1.68]	0.39	[0.25, 0.63] *
	−2500 m	0.38	[0.15, 0.96] *	0.44	[0.23, 0.84] *
	≥2500 m	0.55	[0.30, 1.03]	0.37	[0.25, 0.56] *

Note. Model adjusted for age, education level, living arrangements, subjective economic status, arthralgia or neuralgia, estimated population density, and participation in the other exercise. These data were used as control variables. All independent variables and covariates were included simultaneously. The *p*-value was obtained from multiple logistic regression analysis. OR = odds ratio; CI = confidence interval. * *p* < 0.05.

**Table 3 ijerph-18-11944-t003:** Associations between participation in the square-stepping exercise, interpersonal social networks, and distance to exercise facility.

		Men		Women	
		OR	95% CI	OR	95% CI
Main mode of travel	Car (driving)	1.00		1.00	
	Car (passenger)	0.97	[0.26, 3.64]	0.60	[0.43, 0.84] *
	Cycling or walking	0.43	[0.17, 1.11]	0.82	[0.62, 1.09]
Weak interpersonal social networks	No	1.00		1.00	
	Yes	0.85	[0.33, 2.24]	0.71	[0.39, 1.31]
Distance to exercise facility	≤500 m	1.00		1.00	
	−1000 m	0.56	[0.29, 1.09]	0.61	[0.44, 0.84] *
	−1500 m	0.65	[0.30, 1.41]	0.61	[0.42, 0.88] *
	−2000 m	0.21	[0.06, 0.70] *	0.53	[0.35, 0.82] *
	−2500 m	0.60	[0.25, 1.45]	0.41	[0.25, 0.66] *
	≥2500 m	0.68	[0.28, 1.61]	0.45	[0.28, 0.71] *

Note. Model adjusted for age, education level, living arrangements, subjective economic status, arthralgia or neuralgia, estimated population density, and participation in the other exercise. These data were used as control variables. All independent variables and covariates were included simultaneously. The *p*-value was obtained from multiple logistic regression analysis. OR = odds ratio; CI = confidence interval. * *p* < 0.05.

**Table 4 ijerph-18-11944-t004:** Interactions between sex, interpersonal social networks, and distance to exercise facility, for participation.

	Silver-Rehabili Taisou Exercise		Square-Stepping Exercise	
	OR	95% CI	OR	95% CI
Sex * Car (passenger)	0.74	[0.36, 1.49]	0.41	[0.12, 1.35]
Sex * Cycling or walking	0.65	[0.41, 1.03]	1.63	[0.66, 4.07]
Sex * Weak interpersonal social networks (Yes)	0.90	[0.44, 1.84]	1.02	[0.35, 3.00]
Sex * Distance to exercise facility (>500 m)	0.77	[0.54, 1.11]	0.75	[0.42, 1.34]

Note. Model adjusted for sex, age, education level, living arrangements, subjective economic status, arthralgia or neuralgia, estimated population density, and participation in the other exercise. These data were used as control variables. All independent variables and covariates were included simultaneously. The *p*-value was obtained from multiple logistic regression analysis. OR = odds ratio; CI = confidence interval. * *p* < 0.05.

## Data Availability

The data are not publicly available due to privacy or ethical restrictions. The data that support the findings of this study are available from the corresponding author upon reasonable request.

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
