# Peer review of "Relationships between Participation in Volunteer-Managed Exercises, Distance to Exercise Facilities, and Interpersonal Social Networks in Older Adults: A Cross-Sectional Study in Japan"

_ijerph, 2021, doi:10.3390/ijerph182211944_

Round 1
Reviewer 1 Report
Dear Authors,
even though your results are driven from a limited socio-geographical area I find your work well structured and presented, and I think it can be of interest for stakeholders.
Some improvements are needed to clarify few methodology issues.
In particular, I haven’t understood how you measured the rate of participation in another exercise. I haven’t seen a question on participation in other forms of exercise included in the survey. The lack of this information can be an important bias in the interpretation of the results, you should comment this aspect in your manuscript. If the rate of participation in another exercise was not measured, you should demonstrate that it does not hinder your conclusions.

Author Response
We appreciate all your comments and suggestions. Our responses to your specific comments are below. Revised parts in the main manuscript are highlighted using yellow marker.
Comment 1
In particular, I haven’t understood how you measured the rate of participation in another exercise. I haven’t seen a question on participation in other forms of exercise included in the survey. The lack of this information can be an important bias in the interpretation of the results, you should comment this aspect in your manuscript. If the rate of participation in another exercise was not measured, you should demonstrate that it does not hinder your conclusions.
L101-102
Did you ask questions on participation in other physical activities?
Response 1
In this study, we did not research participation in any exercise other than SRTE and SSE. Therefore, we agree with the reviewers that this is a limitation of this study. We have added to the limitations of this study that although we have adjusted for participation in SSE or SRTE in multivariate analysis, we have not controlled for other types of exercises. (L304–307)
Comment 2
L60
Could you please clarify how you assesses this gap before you gathered your results?
Response 2
To participate in an activity, there are two steps: first, becoming aware of the activity, and second, participating in the activity. To evaluate the gap between awareness and participation, and develop effective strategies to encourage participation, it is necessary to identify the factors related to participation among people who are aware of the activity. As a preliminary step, we have added a report on factors related to the awareness of the activities we conducted. (L66–70)
Comment 3
L111
Could you explain why public transport was not included?
Response 3
In Kasama City and other rural areas in Japan, many people travel by personal car and the use of public transportation is low (239 times / 5,473 trips, 4.4%), so it was not surveyed in this study. Therefore, the absence of variables related to the use of public transportation is not expected to have a significant impact on the results of this study.
Comment 4
L124
Can you demonstrate that 3 questions only can sufficiently describe people social network?
Response 4
Although it is not enough, this study covers many items. For this reason, the number of items was kept to a minimum in consideration of the burden on the research participants. The possibility that social network was not sufficiently investigated has been added to the limitations. (L307–309)
Comment 5
L178
How did you measure the rate of participation in another exercise? Was this question included in the survey?
Response 5
This sentence has been revised as it was initially difficult to understand. (L196–198)
Comment 6
L178
Could you clarify what are you comparing here? Rate of participation was higher than what else?
Response 6
The sentence has been modified for clarity. (L196–198)
Reviewer 2 Report
Dear authors:
First of all I would like to congratulate you on your work and deliver a scientific research manuscript. Here are the points on which your publication should be improved:
- The figure of volunteers performing physical exercise programmes is not understood. Maybe in your home country it makes sense, but in other countries or continents this expression is not understood. Normally it is usually physical trainers, physiotherapists or occupational therapists who are responsible for this work, please explain the creation of this figure of volunteers because from here if this nomenclature is not understood the manuscript makes no sense, moreover with lines 45 and 46 should be explained more clearly.
- The introduction is very brief and you should describe incidence and prevalence of the objective of your study, what do I want to improve? quality of life, physical activity, obesity?
- In material and methods you talk about surveys from 2013. What do they have to do with the year 2021, is it a prospective study? Are you doing a study of the last 8 years in these individuals? It is not clear, there is room for improvement and if they only refer to the year 2013, it is an obsolete study, they should update it.
- They always refer to older adults, but the minimum and maximum age of the participants should be stated. Inclusion and exclusion criteria.
- In the discussion, physical and social variables are mixed, and the objective of the study is not made clear.
- The bibliography is very poor and with outdated references, the most current reference is from 2018. Please expand and update.
The manuscript must improve all these points in order to be accepted and they must clarify the objective of their study, as well as making it clear whether the study was in 2013 or not.
Best regards and thank you very much.
Author Response
We appreciate all your comments and suggestions. Our responses to your specific comments are below. Revised parts in the main manuscript are highlighted using yellow marker.
Comment 1
The figure of volunteers performing physical exercise programmes is not understood. Maybe in your home country it makes sense, but in other countries or continents this expression is not understood. Normally it is usually physical trainers, physiotherapists or occupational therapists who are responsible for this work, please explain the creation of this figure of volunteers because from here if this nomenclature is not understood the manuscript makes no sense, moreover with lines 45 and 46 should be explained more clearly.
Response 1
As you suggest, volunteer-led exercise programmes outside Japan may not be popular. However, due to its lower cost than professional programmes and accessibility to older people in the community, volunteer-managed exercise programmes have received increasing attention in other countries in recent years. I have added a bibliography on this. (L47–51, L103–106)
Comment 2
The introduction is very brief and you should describe incidence and prevalence of the objective of your study, what do I want to improve? quality of life, physical activity, obesity?
Response 2
This study aimed identify factors related to participation and not to identify effects. For this reason, the effects are described in terms of the general benefits of the exercise.
Comment 3
In material and methods you talk about surveys from 2013. What do they have to do with the year 2021, is it a prospective study? Are you doing a study of the last 8 years in these individuals? It is not clear, there is room for improvement and if they only refer to the year 2013, it is an obsolete study, they should update it.
Response 3
The 2013 survey is a baseline survey for the 2021 survey, which considers death and need for care as outcomes. This study uses only data from the 2013 survey. As we were involved in other projects, we were unable to compile the results of our analysis sooner. In addition, geocoding, which used to be inaccurate (block-level), has recently become more accurate, making it possible to analyse with building-level accuracy, necessitating a major update of location information. Although the data is old, we believe that the topics are still relevant.
Comment 4
They always refer to older adults, but the minimum and maximum age of the participants should be stated. Inclusion and exclusion criteria.
Response 4
This survey was conducted on all older people aged 65 and over living in Kasama City who were not certified as having care needs (L88–89). The oldest responder was 101 years old, and the oldest person included in the analysis was 95 years old. (L88–89, L174–175)
Comment 5
In the discussion, physical and social variables are mixed, and the objective of the study is not made clear.
Response 5
In previous studies, it has been reported that participation (≒ accessibility) is related to physical factors such as distance and transportation, and social factors such as social networks. To clarify the purpose of this study, ’accessibility‘ was replaced with ’distance‘ and the hypothesis (L80–82) has been added. (Title, L23, L53, L59, L64, L79, L81, L123, L126, L168, L210, L218, L249, L261, L265, L266)
Comment 6
The bibliography is very poor and with outdated references, the most current reference is from 2018. Please expand and update.
Response 6
The references have been reviewed and updated with the most recent references possible. (References 3, 8)
Round 2
Reviewer 1 Report
Dear Authors,
I congratulate you on responding to all my comments. I confirm my previous opinion that your results can be a starting point reference for stakeholders.
Author Response
Thank you very much for taking the time to review our revised manuscript. We are glad that we could respond to your comments appropriately.
Reviewer 2 Report
Dear authors:
Thank you for correcting and adding the comments I showed you in the first revision of the manuscript. On my part, everything I contributed has been corrected. From my point of view there is a very strong problem in this article and that is that we are talking about a study carried out in 2013, therefore outdated and perhaps does not reflect the current problems and situation, 8 years in a population (and also older) is a significant difference.
Greetings.
Author Response
Comment
From my point of view there is a very strong problem in this article and that is that we are talking about a study carried out in 2013, therefore outdated and perhaps does not reflect the current problems and situation, 8 years in a population (and also older) is a significant difference.
Response
As you pointed out, it is possible that the situation of older people has changed over the past eight years. Accordingly, we have added this to the limitations. (L307-308)